# Antipsychotics in the Management of Disruptive Behavior Disorders in Children and Adolescents: An Update and Critical Review

**DOI:** 10.3390/biomedicines10112818

**Published:** 2022-11-04

**Authors:** Ravi Philip Rajkumar

**Affiliations:** Department of Psychiatry, Jawaharlal Institute of Postgraduate Medical Education and Research (JIPMER), Puducherry 605 006, India; jd0422@jipmer.ac.in; Tel.: +91-413-229-6280

**Keywords:** disruptive behavior disorders, conduct disorder, oppositional defiant disorder, aggression, atypical antipsychotics, risperidone, clozapine

## Abstract

Disruptive behaviour disorders (DBDs) in childhood include conduct disorder (CD) and oppositional defiant disorder (ODD). Though psychological therapies are considered to be the first-line treatment for DBDs, many patients require adjunctive pharmacotherapy for the control of specific symptoms, such as aggression. Three prior systematic reviews have examined the evidence for the use of antipsychotics in DBDs and have concluded that their efficacy is marginal and limited by adverse effects. This paper has two objectives: (i) to summarize the findings of existing systematic reviews of antipsychotics for the management of DBDs in children and adolescents (2012–2017), and (ii) to provide an update to these reviews by examining recent clinical trials of antipsychotics in this population, published in the period from 2 January 2017 to 10 October 2022. The PubMed, Scopus and ScienceDirect databases were searched for relevant citations using the search terms “disruptive behaviour disorder”, “oppositional defiant disorder”, “conduct disorder” and their variants, along with “antipsychotic”, “atypical antipsychotic” and the generic names of all currently approved atypical antipsychotics. Six relevant trials were identified during this period, including five randomized controlled trials and one naturalistic open-label trial. These trials were critically evaluated in terms of outcome measures, efficacy and safety. Overall, the data from these trials suggests that of all available antipsychotics, risperidone appears to be effective in the short-term management of DBDs. All available antipsychotics are associated with significant metabolic adverse effects in this population. These results are discussed in the light of global trends towards increasing off-label prescription of antipsychotic medication in children and adolescents and of recent literature on the neuropharmacology of aggression in this patient population. The need for rational, short-term use of these drugs is highlighted, as well as the importance of post-marketing surveillance for long-term or severe adverse events.

## 1. Introduction

Disruptive behavior disorders of childhood and adolescence (DBDs) include oppositional defiant disorder (ODD) and conduct disorder (CD) [1,2,3]. Other conditions previously included in this group are attention-deficit/hyperactivity disorder (ADHD) and intermittent explosive disorder (IED). However, ADHD is now understood as a distinct disorder of neurodevelopmental origin [4], while IED, which has its onset in adolescence or adult life, is classified with the impulse control disorders [5,6]. These latter conditions may be co-morbid with DBDs but are considered to be distinct from them [7,8]. These conditions are characterized by persistent irritability, disregard for social norms, and recurrent aggression. CD is the more severe subtype of DBD, characterized by unconcern for the rights of others, rule-breaking, aggression and other forms of dissocial behavior, such as theft or truancy. ODD is generally milder in severity and more circumscribed in scope than CD, and it is characterized by defiant and hostile patterns of behavior towards authority figures, temper tantrums and less overt aggression. However, the boundary between these two conditions is fluid, and many children initially diagnosed with ODD may later qualify for a diagnosis of CD. To qualify for a diagnosis of DBD, such patterns of behavior must be persistent and severe and must occur in multiple settings; occasional episodes of defiance or aggression are frequently seen in normal children and do not justify a diagnosis of DBD by themselves [1,2,3]. DBDs are of considerable clinical and public health importance, as they are associated with a number of significant adverse outcomes in adulthood, including mood disorders, adult antisocial personality disorder, substance abuse and dependence, premature discontinuation of education, unemployment, unstable relationships and criminality [9,10,11].

The etiology of DBDs is currently conceptualized in terms of gene–environment interactions, resulting in disturbed emotion regulation and abnormal fear processing. Emotional dysregulation, which reflects altered prefrontal cortical functioning, is linked to symptoms of irritability, while abnormal fear processing, which is related to altered functioning of limbic structures, such as the amygdala, is linked to symptoms of unconcern for others and violation of rules and social norms [12]. Genetic factors that have been linked to DBDs include functional polymorphisms of the genes encoding the enzymes monoamine oxidase A (*MAOA*) and catechol O-methyltransferase (*COMT*), the type 2 and 4 dopamine receptor (*DRD2* and *DRD4*), the serotonin transporter (*SLC6A4*) and—more recently—the genes encoding brain-derived neurotrophic factor (*BDNF)* and the oxytocin receptor (*OXTR*) [13,14]. These loci do not act as isolated risk factors; rather, they influence the vulnerability of a given child to various forms of environmental adversity, including physical and sexual abuse, maternal depression, paternal criminality, harsh parenting practices, and socioeconomic disadvantage [15].

Psychosocial interventions are considered to be the most effective form of treatment for DBDs, particularly when these include multiple components and involve both the child and their parents [16]. However, the availability of these interventions remains a matter of concern; it has been observed that even in high-income countries, up to 70% of children and adolescents do not receive these interventions when needed [17]. In addition, several of the familial and social factors that play a role in the pathogenesis of DBDs can themselves interfere with engagement and retention in psychosocial interventions. These include economic deprivation, parental unemployment, parental criminality and parental substance abuse [18,19].

The discovery of the first antipsychotic medications in the 1950s, beginning with chlorpromazine in 1952, led to a paradigm shift in the practice of clinical psychiatry [20,21]. The serendipitous discovery that chlorpromazine reduced symptom severity in patients with schizophrenia led to a drastic change in the way in which this disorder was understood by the medical profession, leading to a reduction in long-term hospitalizations and a shift towards biological theories of schizophrenia [22]. At first, the mechanism of action of these drugs was not well understood, leading them to be described as “major tranquilizers” or “neuroleptics”. Subsequently, it was found that the efficacy of these medications appeared to correlate with their antagonist potency at dopamine receptors, and most particularly the type 2 dopamine receptor (D_2_). In addition to relieving psychotic symptoms in a significant number of patients, antipsychotic drugs were soon found to reduce the agitation and aggression associated with other psychiatric conditions, such as acute mania, personality disorders and substance-induced behavioral disorders [23,24,25].

As a result of these findings, several tentative attempts were made to examine the efficacy of these medications in childhood disruptive behavior disorders (DBDs), dating back to 1955. Initial reports examined the effect of antipsychotics, such as fluphenazine, trifluoperazine and chlorprothixene, on children and adolescents with DBDs, either with or without comorbid intellectual disability (ID) [26,27,28]. Subsequently, small clinical trials examined the effects of more high-potency antipsychotics, such as haloperidol, in this population [29]. Although these publications were largely of an anecdotal nature, interest in the use of antipsychotics in DBD received further impetus from research on genetic risk factors for this group of disorders. Most of the vulnerability alleles identified in this research were related to monoaminergic neurotransmission, particularly involving the dopaminergic and serotonergic systems [14,15,30]. As these systems are the locus of the pharmacological actions of most antipsychotic medications, it is biologically plausible that antipsychotics may alleviate some of the clinical manifestations of DBDs. Atypical antipsychotics, which are characterized by selective antagonism of both dopamine and serotonin receptor subtypes [31], represent a theoretically attractive treatment option. The first controlled clinical trial of an atypical antipsychotic for the symptomatic management of DBDs was conducted in 2000, involving children with DBDs and comorbid attention-deficit/hyperactivity disorder (ADHD). The subsequent decades saw a marked increase in the off-label use of atypical antipsychotics for the treatment of specific DBD symptoms, such as aggression and impulsivity; however, relatively little was known about the long-term efficacy and safety of this treatment approach [32,33,34]. 

A meta-analysis conducted in the year 1999 found that only three studies, all involving children with both ID and DBD, fulfilled the criteria for inclusion, and none of these studies provided firm evidence for the use of typical antipsychotics in the symptomatic management of this patient group [35]. Subsequent trials suggested that these drugs were not superior to placebos in the management of DBD in patients with comorbid ID [36,37,38], and experts in the fields highlighted the absence of supporting evidence for the use of these drugs in DBDs, as well as the ethical issues associated with their widespread prescription [39,40]. To address these gaps in the evidence base, three systematic reviews were carried out in the period 2012–2017, each of which examined the evidence for the efficacy of antipsychotics, particularly atypical antipsychotics, in this population [41,42,43]. Subsequent to the publication of these reviews, several clinical trials evaluating the safety or efficacy of antipsychotics in childhood DBDs were published in the period 2017–2022, and off-label prescription of antipsychotics in this population remained high [44].

The objectives of the current review are:
To summarize the findings of existing systematic reviews, evaluating the efficacy of antipsychotics in children and adolescents with DBDs;To update these existing reviews by critically evaluating recent (2017–2022) clinical trials of antipsychotics in this population, in terms of both efficacy and safety.

## 2. Materials and Methods

### 2.1. Identification and Summarization of Existing Systematic Reviews

To address the first objective of this review, the PubMed/MEDLINE, Scopus, ScienceDirect and Cochrane databases were searched for relevant systematic reviews using the terms “disruptive behavior disorder”, “childhood disruptive behavior disorder”, “conduct disorder”, “oppositional defiant disorder” and their alternative spellings and plural forms, in conjunction with “antipsychotic”, “antipsychotics”, “atypical antipsychotic” or “atypical antipsychotics”. A total of three systematic reviews were identified using this strategy, published in 2012, 2015 and 2017, respectively [41,42,43]. These reviews were summarized under the following headings: number of trials included, number of subjects studied, drug(s) evaluated in the included trials and results pertaining to efficacy and safety. These reviews identified certain key limitations of the existing evidence base, which were tabulated and organized under five main headings.

### 2.2. Identification of Recent Clinical Trials of Antipsychotics in Children and Adolescents with DBDs

The second objective of this review was to identify recent clinical trials of antipsychotics in children and adolescents with DBDs and to evaluate the data on efficacy and safety that could be obtained from these trials. For this purpose, a comprehensive literature search was conducted, involving all controlled clinical trials of antipsychotics published in the period from 2 January 2017 to date (10 October 2022). The date 2 January 2017 was selected as the starting point because the most recent review included all papers published up to 1 January 2017. The PubMed/MEDLINE, Scopus and ScienceDirect databases were searched for all relevant clinical trials involving antipsychotic therapy, either alone or as an adjunctive treatment, in children and adolescents with DBDs.

The search strategy involved the following terms: “disruptive behavior disorder”, “disruptive behavior disorders”, “childhood disruptive behavior disorder”, “oppositional defiant disorder”, “conduct disorder” and “conduct disorders”, in conjunction with “antipsychotic”, “antipsychotics”, both alone and in combination with “typical” or “atypical”, as well as the generic names of all the antipsychotics covered by the most recent review and those recently approved for the treatment of other psychiatric disorders (“amisulpride”, “aripiprazole”, “asenapine”, “clozapine”, “lurasidone” “olanzapine”, “paliperidone”, “quetiapine”, “risperidone”, “sertindole”, “ziprasidone”, “zotepine”).

Of a total of 290 unique citations received using these search terms, 243 were excluded based on the title and abstract. The full texts of the remaining 47 papers were examined. Studies were included only if they fulfilled the following criteria:(a)Prospective clinical trials (i.e., no retrospective chart reviews or case series);(b)Studies involving children or adolescents (age 0–18);(c)Studies involving patients with a diagnosis of DBD (ODD or CD), with or without comorbid ID or ADHD;(d)Clear reporting of outcomes (efficacy, adverse events or both) using a standardized measure, such as a valid rating scale or equivalent instrument.

The reference lists of the studies retrieved using this method were checked for further citations of interest, but no trial fulfilling the above inclusion criteria was identified through this method. A total of six clinical trials fulfilling the above criteria were included in the current review [45,46,47,48,49,50].

### 2.3. Identification of Unpublished Trial Data

As earlier systematic reviews had identified relevant results from an unpublished trial, the ClinicalTrials.gov database was searched to examine if any unpublished trials of relevance could be identified for the period 2017–22, using the diagnostic categories “Disruptive Behavior Disorder” and “Childhood Disruptive Behavior Disorder” (provided by the database’s classificatory system), along with “antipsychotic” and “antipsychotics”, both alone and in combination with “typical” or “atypical”, as well as the list of antipsychotics mentioned above. Of 27 records retrieved through this search, only one study of potential relevance was identified in the time period 2017–22. This trial (identifier: NCT02063945) was a head-to-head, open-label comparison of risperidone and methylphenidate in children and adolescents with DBDs and comorbid ADHD. Only five subjects were recruited for the study, and it was prematurely terminated as of 2020 due to difficulty in recruiting participants; no results have been posted to date. Therefore, it was not possible to include the detailed results of this abandoned trial in the current review [51].

### 2.4. Qualitative Synthesis of Included Trials

For each trial, information was tabulated as follows: year of publication; study inclusion criteria, including comorbid diagnoses permitted, sample size and age distribution; study design; study interventions (drug, dosage and whether used as monotherapy or adjunctive therapy); and study results pertaining to efficacy and safety, including the frequency of reported adverse events. 

For all randomized controlled trials, study quality was assessed using the Jadad scale [52], which is a well-established and valid method to assess the quality of reporting in such trials [53]. The overall results of these recent studies were then examined in light of the concerns identified in earlier systematic reviews.

The review process is depicted graphically in Figure 1.

## 3. Results

### 3.1. Systematic Reviews of Antipsychotic Therapy for DBDs

Three earlier systematic reviews were identified and included in the current review [41,42,43]. The key findings of these reviews are summarized in Table 1.

In the first of these reviews (2012), Pringsheim and Gorman examined the results of eight controlled clinical trials of antipsychotics in the management of DBDs, either with or without comorbid ADHD or ID [41]. This review included only trials of atypical antipsychotics, covering a total of 640 children and adolescents (mean sample size = 80, range, 13–335). Seven of these eight trials involved risperidone, administered at a mean dose of 0.25 to 2.9 mg/day. Five of the seven trials of risperidone found that this drug was superior to the placebo in terms of primary outcomes, usually defined as a reduction in scores on a standard rating scale of DBD symptomatology or aggression; the other two found no significant difference on primary outcomes, though both of these had small sample sizes. All the five trials reporting positive outcomes involved children with ID. A single trial examined the efficacy of quetiapine (mean dose 294 mg/day) in a small sample of adolescents and found that this drug had a small but significant impact on symptoms of conduct disorder (mean decrease in symptom scores of 2.5 with quetiapine vs. 0.5 with placebo). Most of these trials were short in duration, involving a study period of 4–10 weeks. Only one trial, involving adolescents with ID and DBD, examined the efficacy of risperidone (0.25–1.5 mg/day) over a period of 6 months. This study involved an inherent bias, as it recruited only prior responders to risperidone treatment; these subjects were randomized to receive either risperidone or placebo as continuation treatment. It was found that the symptom recurrence rate was significantly higher for the placebo (42%) than for risperidone (27%). The authors of this review noted several limitations of the available data, such as underreporting of adverse effects, a lack of research that was not industry-funded, and a lack of long-term data on efficacy; they also highlighted the need for concurrent psychosocial interventions.

The second review was conducted in 2015 by Pringsheim et al. [42] and included studies of both typical and atypical antipsychotics, covering a total of 11 trials involving 896 children and adolescents (mean sample size = 81, range, 13–335). This review also included a meta-analysis of efficacy. Eight of the trials included in this review (seven of risperidone and one of quetiapine) were the same as those included in the 2012 review and have been discussed above. The three additional trials covered in this review assessed the use of haloperidol, thioridazine and risperidone. In the first of these, children with CD were randomized to receive either haloperidol (1–6 mg/day), lithium or placebo for 4 weeks. Both haloperidol and lithium were superior to the placebo in reducing CD symptoms, but haloperidol was poorly tolerated, compared to lithium. This study was rated as being of “very low” quality by the reviewers, and it was noted that the magnitude of the treatment effects was not reported. In the second, 27 children with ID and comorbid ADHD or CD were randomized to receive either thioridazine (1.5 mg/kg), methylphenidate (0.4 mg/kg) or a placebo for a period of three weeks. Though the study reported that thioridazine was superior to the placebo in reducing CD symptoms, this was not the primary outcome of the trial; the study was primarily designed to assess the effects of these drugs on cognitive and motor performance. This trial was also rated as being of “very low quality”. In the third, adjunctive risperidone (mean dose 1.65 mg/day) was compared to a placebo in a sample of children with ADHD and comorbid DBD, already receiving stimulant therapy and psychosocial interventions. Over a period of 9 weeks, risperidone was found to be superior to the placebo in reducing ODD symptoms and aggression towards peers but not in reducing ADHD or CD symptoms. Synthesizing the available evidence, the authors concluded that there was moderate-to-good evidence for the short-term use of risperidone in reducing the disruptive and aggressive behavior associated with DBDs, regardless of the presence of comorbid ID or ADHD. However, they also noted the need for better data regarding drug safety, and the lack of significant evidence for other antipsychotics or for longer-term treatment. They also highlighted the problem posed by unpublished studies that may have yielded negative results, thus, leading to an over-estimation of the efficacy of antipsychotics.

Despite the limitations of the available data, off-label use of antipsychotics in this patient population continued to increase significantly in the decade between 2005–2014 [34,44]. To address this, a subsequent systematic review and meta-analysis, published as part of the Cochrane Database of Systematic Reviews, was conducted by Loy et al. in 2017 [43]. This paper represents the most recent synthesis of data on the efficacy and safety of antipsychotics in the management of DBDs in the available literature. This review included a total of 10 trials covering 896 children (mean sample size = 90, range, 13–335). Of these, the eight trials of risperidone and the single trial of quetiapine have already been described as part of the preceding two reviews. The single additional study included in this review was a controlled clinical trial, comparing the atypical antipsychotic ziprasidone (20–40 mg/day) with a placebo over a period of 9 weeks in children with DBD and an IQ of 55 or above (indicating mild or no ID). At study termination, no significant difference was identified between ziprasidone and the placebo either in terms of efficacy or adverse effects. It is notable that this study, although conducted in 2011, has not been published in a peer-reviewed journal to date, and the authors of the review had to base their analysis on unpublished data obtained through a direct request to the researchers. The authors concluded that there was low-to-moderate quality evidence for the use of risperidone in treating aggression and conduct problems in DBDs but no adequate evidence for the use of any other antipsychotic. This review also included an in-depth coverage of safety and tolerability issues. It was found that patients receiving risperidone experienced a mean weight gain of 2.37 kg greater than those receiving placebo; in trials where risperidone was administered in combination with stimulants, this figure was slightly lower (2.14 kg) but still significantly higher than the weight gain observed with the placebo. It was also observed that hyperprolactinemia was significantly more likely to occur with antipsychotic treatment than with placebo; however, the reporting of other adverse events was inconsistent across trials and direct comparisons could not be made. The final recommendation made by the review authors was that there was short-term efficacy for the use of risperidone in DBDs in children aged 5 and above but that this treatment should only be administered along with psychosocial interventions and that the risk of weight gain remained a significant clinical concern.

### 3.2. Limitations of the Existing Evidence Identified in Earlier Systematic Reviews

A careful study of these three systematic reviews reveals certain common themes (Table 2). First, the vast majority of the published evidence is related to a single drug—risperidone—suggesting that the off-label use of other antipsychotics in DBDs is unsupported by evidence. Second, there is significant heterogeneity in patient populations (comorbid ID, comorbid ADHD, age groups, concurrent use of other medications or psychosocial interventions), as well as outcome measures, posing significant problems when attempting to compare or synthesize individual study results. Third, the concern regarding unpublished data flagged by Pringsheim et al. [42] proved to be valid, as shown by the fact that one of the clinical trials included in the Cochrane review has not been published to date [43]. Fourth, the reporting of adverse events is generally inconsistent and of low quality across trials. Finally, there is a lack of data on efficacy beyond a period of 4–10 weeks. These limitations were used as the basis for the evaluation of more recent clinical trials.

### 3.3. Recent Clinical Trials of Antipsychotics in the Management of DBDs

A total of six recent clinical trials were identified for inclusion in this review [45,46,47,48,49,50]. The details of these trials are summarized in Table 3.

### 3.4. Study Design and Quality

Of the included studies, four were placebo-controlled trials of adjunctive antipsychotic therapy, one was a randomized controlled trial of antipsychotic monotherapy involving an active comparator and one was a naturalistic trial involving an active comparator but no specific process of randomization and blinding. None of the trials involved children or adolescents with intellectual disability; however, five of six trials (83.3%) included subjects with comorbid ADHD. These six studies covered a total of 461 participants (mean sample size, 77; range, 24–165), spanning an age range of 6–16 years. Trial duration ranged from 6 weeks to 6 months.

For all randomized controlled trials, study quality was assessed using the Jadad scale. One trial could not be rated using this scale, as it was a naturalistic trial that did not involve randomization or blinding. The Jadad score ranged from 3 to 5 for the five randomized controlled clinical trials (mean score = 4), indicating moderate-to-good quality for these studies.

Of the included trials, one was a continuation conducted in patients who had shown a prior good response to antipsychotic medication, raising the possibility of a bias towards positive trial outcomes [46]; on the other hand, the most recent trial included subjects who were still symptomatic after the optimization of other medication, indicating an attempt to minimize bias and include participants with significant symptoms [50]. No other significant sources of bias were identified in the other trials.

### 3.5. Efficacy

Five of the six studies included examined the efficacy of antipsychotic therapy in reducing DBD symptomatology; the remaining study examined safety in terms of effects on cognition as the chief outcome measure and is discussed in Section 3.3. Among these trials, four included children and adolescents with comorbid ADHD and DBD [46,47,49,50], and one included only children with CD and significant levels of aggression [48].

In a single naturalistic study involving only male participants with ADHD and ODD (*n* = 40), monotherapy with risperidone (mean dose 1.5 mg) was compared with the stimulant methylphenidate (20 mg/kg) over a period of 6 months. At the conclusion of the study, both drugs were found to be equally effective in managing symptoms of ODD, but methylphenidate was superior to risperidone in reducing symptoms of inattention and hyperactivity [49].

In two short-term, placebo-controlled studies of comorbid ADHD and DBD, both lasting 8 weeks, adjunctive risperidone (0.5–2.5 mg) was superior to a placebo in the management of specific symptoms—aggression in one trial and oppositional symptoms in the other [47,50]. All subjects in these studies were receiving stimulant therapy with methylphenidate at standard doses. In one of these trials, both risperidone and divalproex were evaluated as active drugs; in this trial, risperidone was superior to both divalproex and a placebo in reducing aggression [40].

In a single continuation trial of prior responders to adjunctive risperidone, involving children with a mean age of 9.2 years studied over a period of 12 weeks, participants randomized to risperidone (mean dose 1.56 mg) did not differ from those receiving placebo on primary outcome measures. However, risperidone appeared to be superior on certain secondary outcome measures, such as positive social behaviour and reactive aggression [46].

Finally, in a single randomized trial of children and adolescents with CD, clozapine (0.6 mg/kg/day) was comparable to risperidone (0.05 mg/kg/day) on the primary outcome measure of overt aggression. Clozapine appeared superior to risperidone on the secondary outcomes of global functioning and delinquent behaviour [48].

### 3.6. Safety and Tolerability

All six trials included in this review reported data on safety and tolerability, including serious adverse events and drop-out rates. Significant weight gain was reported in all but one of the trials of risperidone, ranging from 1.4–2.2 kg at 8 weeks [47,50] to 3.6 kg at 6 months [49]. Over a period of 6 months, it was observed that 25% of participants receiving risperidone experienced an increase in body weight greater than 5%; however, this figure is based on a single trial [49]. In a continuation trial in which all participants had received prior risperidone treatment for 9 weeks, there was no subsequent difference in weight change between risperidone and placebo over a period of 12 weeks [46]. A trial comparing risperidone with clozapine over 4 weeks found a mean weight gain of 4.1 kg with risperidone and 2.8 kg with clozapine; this difference was significant with respect to time but not treatment group, indicating that the observed difference between groups was not significant [48].

Common adverse effects reported with antipsychotics, along with their frequency, were reported in five trials. In one of these, adverse effects were listed but no data on their frequency was provided [47]; in the others, frequencies of each adverse effect were provided in the article text or supplementary material published along with the article [46,48,49,50]. These are summarized in Table 4. Adverse events that were reported to occur more frequently with antipsychotic than with placebo included somnolence or sedation, syncope, extrapyramidal adverse effects (tremors or stiffness), anxiety, constipation, increased appetite, skin rash, menstrual irregularity and nocturnal enuresis. However, the placebo group in these trials was receiving concurrent medication (usually with methylphenidate), making direct comparisons of the frequency of these events difficult.

Serious adverse effects reported in these trials included neutropenia and suspected drug-induced dyskinesia. In a trial comparing adjunctive risperidone, divalproex and placebo, 1 of 18 subjects receiving risperidone (5.6%) developed neutropenia [50]; in a trial comparing risperidone and clozapine, 2 of 12 subjects receiving clozapine (16.7%) developed neutropenia, but this was not observed in any of the subjects receiving risperidone [48]. Suspected dyskinesia severe enough to warrant treatment discontinuation was reported in 1 of 54 (1.8%) participants receiving risperidone over a period of 12 weeks [46].

Changes in laboratory parameters were monitored in five studies [46,47,48,49,50]. Risperidone use was associated with significant elevations in serum prolactin in two trials [46,47]. Risperidone was also associated with elevated total cholesterol when administered over 6 months [49]; however, this effect was not noted in a shorter (8 week) trial [47]. No significant changes were reported in plasma glucose, renal function tests (urea and creatinine) or liver enzymes.

Given the concerns related to cognitive blunting and dulling with the use of older antipsychotics in children [54], a single clinical trial examined the effects of adjunctive risperidone (mean dose 1.7 mg/day), compared to placebo, on two measures of cognition—the Continuous Performance Test-II (CPT-II) and the Digit Span subscale of the Weschler Intelligence Scale for children. Over a period of six weeks, no significant differences were identified between the two groups in terms of cognitive performance on either measure [45]. However, all participants in this study were receiving concurrent methylphenidate, as well as psychosocial intervention in the form of parent training.

An examination of drop-out rates across all included studies revealed that antipsychotic therapy was not associated with a significant increase in study discontinuation, both in general and for specific or severe adverse events.

## 4. Discussion

Standard practice guidelines, such as those published by the United Kingdom’s National Institute for Health and Care Excellence, take a cautious approach to the prescription of antipsychotics for children and adolescents with DBDs, with their use confined to the short-term treatment of severe aggression resistant to psychosocial interventions [55]. Despite such recommendations, the use of atypical antipsychotics in children and adolescents with DBDs has increased markedly in the past two decades [32,33,34,44,56]. Most of the data on antipsychotic prescribing patterns comes from high-income countries, as there is a lack of published data on the use of these drugs among youth from low- and middle-income countries [44]. For example, a study of over 180,000 youth with ADHD found that 2.6% were prescribed antipsychotics in the year following their diagnosis; among those receiving an antipsychotic, almost 48% were not treated with stimulants. In many cases, these drugs were used for the management of comorbid DBDs, particularly ODD [57]. Off-label prescription of antipsychotics for DBDs is also frequent; a study of children aged 2–7 receiving antipsychotics in the United States found that, in the period 2009–2017, the proportion of these children with a diagnosis of ODD or CD rose from 15% to 21% [58]. Similarly, a study of Canadian children found that 44% of prescriptions for risperidone and almost 50% of prescriptions for aripiprazole were for a diagnosis of CD [59]. A study of children (aged 1–17 years) receiving antipsychotic prescriptions found that 19% of the children with DBDs did not receive appropriate psychosocial treatment [60]. In populations at a higher risk of developing DBDs, such as children in foster care, the over-prescription of antipsychotics for children and adolescents with these disorders may be more frequent. In a study of 128 children in foster care, up to 16% were receiving antipsychotics for various indications; of these children, 29% received a DBD diagnosis. Drugs used in these children included aripiprazole, asenapine, paliperidone, olanzapine and risperidone. It was noted that many of these prescriptions did not follow accepted practice guidelines; moreover, none of these drugs, except risperidone, has been evaluated for the management of DBDs in controlled clinical trials [61]. Another issue of concern related to the use of these drugs is the duration of treatment. Though most controlled clinical trials of antipsychotics in children with DBDs are of short duration (4–10 weeks), a study of 316 young children from a lower-income group (age < 6 years) receiving antipsychotics reported a mean duration of treatment of 2.6 years, with 27% having taken these medications for over four years [62]. Finally, despite the fact that all controlled trials of antipsychotics in children involve single drugs, antipsychotic polypharmacy is frequently encountered in real-world settings [62,63]. These concerns have been flagged by the authors of earlier systematic reviews, and though there is some evidence of a gradual decline in antipsychotic prescription in younger children in more recent research [64], the above results highlight the need for more robust evidence on the efficacy and safety of antipsychotics in children and adolescents with DBDs [39,43,65].

It was with the above considerations in mind that the current review was undertaken. The six studies reviewed in this paper represent a significant addition to the literature; they are of generally good methodological quality, provide valuable information on certain unanswered clinical questions and have excluded children with intellectual disabilities in whom aggression may have a distinct neurobiological basis and may not respond well to this drug class [36,37,38,39,66]. With regard to efficacy, two of the reviewed studies provide further support for the short-term use of risperidone in the management of DBDs in children with comorbid ADHD; both these studies were of good quality [47,50]. On the other hand, the maintenance study conducted by Findling et al. suggests that, among children with DBDs and ADHD who have responded to an initial trial of risperidone, this drug can be discontinued over the next three months without a substantial risk of symptomatic worsening [36]. Though the latter result requires replication, it does provide some support for a more time-limited use of these medications in this vulnerable population. These results also highlight the need for appropriate concurrent treatment with stimulants and psychosocial interventions, as these treatments may have contributed substantially to the favorable outcomes described above.

In contrast with these results, the results of the naturalistic study conducted by Masi et al. suggest that antipsychotic monotherapy may be inferior to stimulant therapy, at least in some respects, in the initial management of DBDs with comorbid ADHD [49]. This result, though provisional and based on a less rigorous research protocol, suggests that the initial prescription of antipsychotics following a diagnosis of ADHD is best avoided [57].

Though there is a reasonable level of evidence for the use of antipsychotics in young people with comorbid ADHD and DBDs, there is, as of yet, no substantial evidence for the use of this drug class in DBDs alone. The single relevant trial published in the period 2017–2022 should be interpreted with caution, as it was of a somewhat lower methodological quality and did not include a placebo group [48]. Moreover, one of the drugs used in this trial—clozapine—was associated with emergent neutropenia in 17% of those receiving it, which would be considered an unacceptable level of risk. In addition, both drugs in this trial were associated with high rates of emergent behavioral adverse effects, such as irritability and anxiety. Although it is unclear to what extent these effects were causally related to the drugs used, these results suggest that the conclusions reached by earlier reviewers remain valid, and these drugs should be avoided in the management of CD or ODD pending further evidence.

With regard to safety concerns, weight gain was identified as a consistent adverse effect across trials. Given the potential long-term hazards associated with weight gain, particularly in young people, this remains a matter of concern. Moreover, the mean weight gain reported in the above trials may not reflect the actual risks associated with antipsychotic monotherapy in children or adolescents. Many of the trial participants were receiving concurrent stimulant therapy, which is associated with reduced appetite and possible weight loss, particularly in younger children [67]. The observation that around one quarter of trial participants gained over 5% of their baseline body weight over a longer treatment period is a cause for greater concern, though this figure was based on a single trial and requires replication [49]. Likewise, the elevations in serum cholesterol and prolactin represent a significant adverse effect, as they could contribute to subsequent medical conditions, such as metabolic syndrome [68] or osteoporosis [69], if antipsychotic therapy is administered for a prolonged period. A severe adverse effect of particular concern identified in recent study data was neutropenia, which was specifically associated with clozapine but was also reported in association with a child receiving risperidone therapy [38,40]. Given that younger age is a risk factor for clozapine-induced neutropenia, this drug is best avoided in this population, and careful monitoring of white cell counts would also be warranted if risperidone is being prescribed. The evidence from these short- to medium-term trials suggests that the risk of extrapyramidal adverse effects or cognitive impairment is low or minimal [45,46,47,48]. However, it must be borne in mind that these results were obtained in the context of low-dose antipsychotic use in carefully selected and monitored study populations; it is possible that extrapyramidal or cognitive adverse effects could appear when antipsychotics are prescribed for DBDs at higher doses or longer durations [70]. Adverse effects, such as sedation and enuresis, though not listed as “severe” by study authors, can also cause significant distress and social and academic dysfunction in this age group and should also be taken into account when prescribing antipsychotics to children with DBDs [71,72].

An important limitation of earlier research in this field is that it was largely based on trials conducted in high-income countries. In contrast, two of the studies included in this review were from middle-income countries (Iran and Mexico). This is a welcome trend, as there is an urgent need to examine the safety and efficacy of these drugs in ethnically diverse populations. Given that antipsychotics are often used “off-label” in these countries, sometimes in formulations that are not approved in this age group [62], the results of this research should stimulate more rational prescribing practices in these settings [55].

When examining the results of this updated review in the light of the five concerns raised in Table 2, a mixed picture emerges. The first point—the preponderance of risperidone as the study drug in clinical trials, remains a significant concern. The only other antipsychotic evaluated was clozapine, which was associated with significant safety concerns. Though several others antipsychotic drugs (aripiprazole, olanzapine, quetiapine, ziprasidone) are used off-label for the management of DBDs, the available literature still does not permit any clear recommendations to be made regarding their safety and efficacy. The current state of the literature in this field highlights the urgent need for ethically and methodologically sound, controlled clinical trials of other antipsychotics in children and adolescents with DBDs, covering both shorter and longer treatment durations and evaluating both safety and efficacy. Regarding the confounding effects of comorbidity (point 2), the more recent trials included in this review addressed this issue partially—children with intellectual disability were excluded, but most trial participants still had comorbid ADHD. When considering publication bias (point 3), an examination of unpublished trials revealed only a single unpublished study of risperidone, which was prematurely terminated due to difficulties in subject recruitment and involved only five subjects at the time of termination; it is unlikely that the non-inclusion of this trial was a significant source of bias. When compared with earlier trials, the reporting of adverse events (point 4) was of superior quality in recent trials, and one of the published reports was solely concerned with cognitive adverse events. Finally, the duration of clinical trials (point 5) remained a matter of concern, with no trial lasting longer than 6 months; however, a positive finding in this respect was the publication of one discontinuation trial, which suggested that antipsychotic discontinuation was not associated with significant deterioration following successful short-term treatment. It can be concluded that the recent literature in this area does represent a meaningful improvement over earlier research in two of the five aspects, and a partial improvement in the remaining three.

It is possible that further breakthroughs in this field may come not just from improvements in trial methodology but also from advances in three fields of research: the neurobiology of disruptive behavior disorders, the pharmacogenomics of antipsychotic response and adverse events, and the availability of antipsychotics acting through novel molecular mechanisms. Understanding the molecular mechanisms associated with DBD symptomatology—both dopaminergic and other targets, such as oxytocin, the oxytocin receptor and diverse serotonin receptor subtypes (5-HT_1B_, 5-HT_3_)—could lead to the development of alternative pharmacotherapies or more rational use of currently available antipsychotics [9,10]. For example, antipsychotics that act at the aforementioned serotonin receptors, such as asenapine or quetiapine, could represent alternatives to risperidone. Pharmacogenomic methods could also be used to identify children or adolescents who might respond better to antipsychotic therapy, or who are at a higher risk of specific adverse effects. There is preliminary evidence of the potential utility of this approach in children and adolescents with other behavioral disorders [73,74]. Changes in the methylation of specific genes linked to these pathways, which have been identified in DBDs, could also serve as potential biomarkers of treatment response, though this approach has not yet been fully evaluated in young people [75]. Finally, antipsychotics acting through non-dopaminergic and non-serotonergic mechanisms have recently been evaluated and found effective in the management of adults with psychotic disorders. These include drugs acting through glutamatergic and cholinergic mechanisms [76]. As both of these pathways have been associated with specific aspects of DBD symptomatology [77,78], it is possible that these novel drugs may offer advantages in safety and efficacy over commonly used antipsychotics.

The current review is subject to certain limitations. Apart from those already discussed earlier, these include: (a) the lack of studies examining specific dimensions or more precise phenotypes of DBD symptomatology, (b) the remaining possibility of missing data from unpublished trials, (c) the absence of pooled outcome measures of drug efficacy, due to the heterogeneity in the included trials, (d) the paucity of data on behavioral adverse effects, when compared with the availability of data on metabolic adverse events, (e) the absence of clinical trials comparing antipsychotics with more evidence-based forms of treatment, such as parent training or school-based interventions, and (f) the lack of data on predictors of response, whether biological (e.g., genotype or epigenetic changes) or psychosocial (e.g., family environment, socioeconomic status, or history of abuse or neglect).

## 5. Conclusions

The history of antipsychotic usage for the management of DBDs in childhood and adolescence is almost as old as that of antipsychotics themselves, spanning the period from 1955 to date. However, it is only in the past two decades (1999–2022) that randomized controlled trials of good quality have been conducted to evaluate the safety and efficacy of these drugs. Recent clinical trials of antipsychotics in children and adolescents with DBDs represent a slight increase in study quality over earlier research. The results of these trials support earlier trial data and recommendations on the use of risperidone in the specific clinical scenario of DBD symptoms in children and adolescents with comorbid ADHD and also highlight the attendant risks of weight gain and prolactin elevation. Despite forty years of clinical trials in this area, there is, as of yet, no sufficient evidence to recommend the use of antipsychotics in DBDs in general. The findings of this review will be of use to researchers involved in the design of subsequent interventional studies. It is also hoped that the evidence summarized in this paper will serve to guide to clinicians towards more rational, short-term prescribing of these drugs and to pay due attention to psychosocial factors and their management in children and adolescents with these challenging behavioral disorders.

## Figures and Tables

**Figure 1 biomedicines-10-02818-f001:**
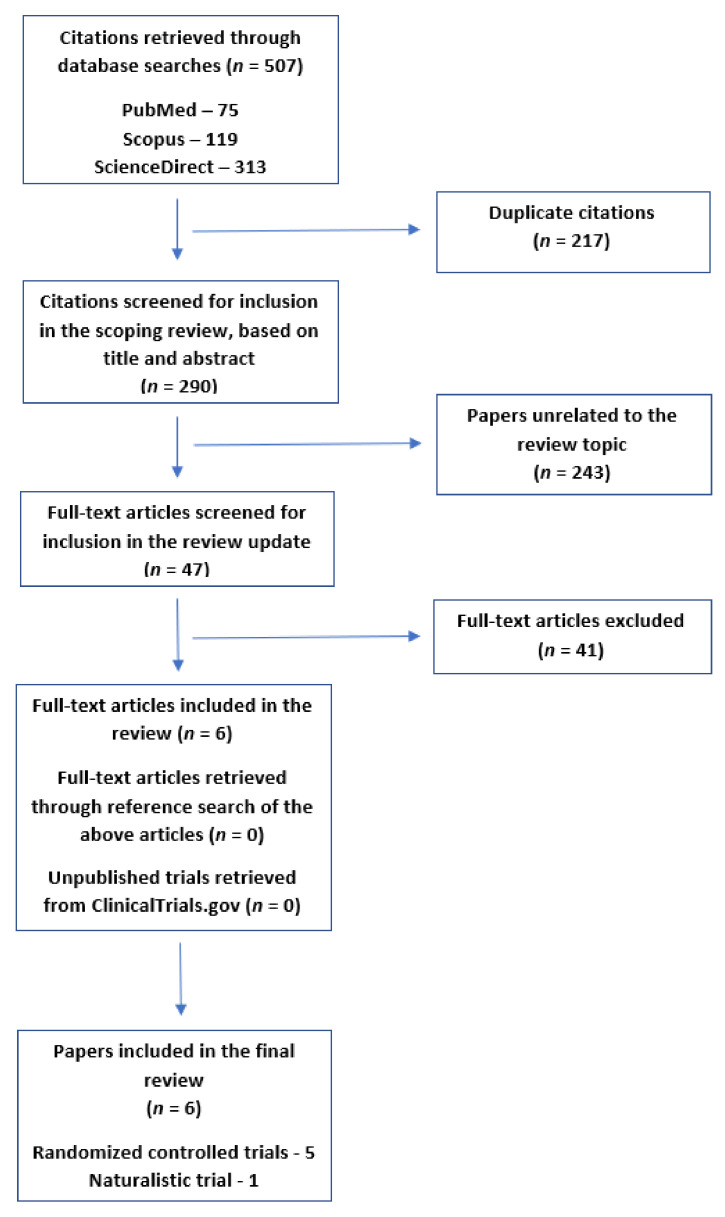
Flow diagram depicting the selection of articles for the current review.

**Table 1 biomedicines-10-02818-t001:** Earlier systematic reviews of clinical trials of antipsychotics in children and adolescents with DBDs.

Review	Year of Publication	Number of Trials Reviewed	Drugs Included	Number of Patients Included	Results	Comments
Pringsheim and Gorman [31]	2012	8	Risperidone (7), quetiapine (1)	640	Risperidone > placebo (0.25–2.9 mg/day) for DBD symptoms in 5 of 7 trials; quetiapine (294 mg/day) > placebo for CD symptoms in 1 trial.	Only atypical antipsychotics included; reviewers noted underreporting of adverse effects, industry funding of all studies, short-term duration of trials.
Pringsheim et al. [32]	2015	11	Risperidone (8), quetiapine (1), haloperidol (1), thioridazine (1)	896	Moderate-to-good quality evidence for short-term risperidone in managing DBD symptoms; insufficient or low-quality evidence for other drugs.	All antipsychotics included; reviewers noted the need for head-to-head comparisons of different medication and of medication with psychosocial therapies, as well as the need for long-term trials and discontinuation studies.
Loy et al. [33]	2017	10	Risperidone (8), quetiapine (1), ziprasidone (1)	896	Low-to-moderate quality evidence for short-term risperidone in managing DBD symptoms; insufficient evidence for other drugs.	Reviewers noted significant weight gain and elevated prolactin with risperidone; recommended use only in children aged 6 and above, with concurrent psychosocial therapies and weight monitoring.

**Table 2 biomedicines-10-02818-t002:** Limitations of existing clinical trials of antipsychotics in DBDs.

Limitation	Consequences
The majority of controlled clinical trials involve a single antipsychotic, namely risperidone.	Inadequate evidence or rationale for the prescription of other antipsychotics in DBDs, though this practice is frequent in real-world settings.
Heterogeneity in patient populations, including age distribution, gender, comorbid ID and/or ADHD.	Uncertainty related to the effect of antipsychotics on DBDs per se, as opposed to effects on ADHD; lack of evidence on the use of antipsychotic treatment for DBDs without comorbidity.
Publication bias may lead to the non-publication of antipsychotic trials in DBDs with negative results (as observed with ziprasidone).	Over-estimation of the beneficial effects of antipsychotics on DBDs; lack of access to valuable data on safety and efficacy measures.
Inconsistent or poor reporting of adverse events.	Lack of adequate data on safety to guide clinicians, as well as patients and caregivers.
Short-term nature of most clinical trials.	Inadequate evidence regarding the safety or efficacy of antipsychotics beyond 8–10 weeks, though use for months or years is frequent in real-world settings; lack of evidence on when and how to discontinue antipsychotics in DBDs.

Abbreviations: ADHD, attention-deficit/hyperactivity disorder; DBD, disruptive behavior disorder; ID, intellectual disability.

**Table 3 biomedicines-10-02818-t003:** Clinical trials of antipsychotics in children and adolescents with disruptive behavior disorders included in the current review (2017–22).

Study Details	Study Design	Sample Characteristics	Intervention	Duration	Primary Outcome Measure	Results	Jadad Score
Farmer et al., 2017 [45]	Randomized, controlled, double-blinded	Children aged 6–12; DBD with comorbid ADHD; presence of severe physical aggression; IQ > 70 (*n* = 165)	Adjunctive risperidone (mean dose 1.7 mg/day) vs. placebo; all patients received ongoing methylphenidate and parent training.	6 weeks	Cognitive performance as assessed by CPT-II and Digit Span Subscale of Weschler Intelligence Scale (Childhood Version)	No significant difference in CPT-II or Digit Span performance between groups.	3
Findling et al., 2017 [46]	Randomized, controlled, double-blinded extension	Children aged 6–12; DBD with comorbid ADHD; presence of severe disruptive behavior; IQ > 70; previous good acute response to risperidone (*n* = 103)	Maintenance risperidone (mean dose 1.56 mg/day) vs. placebo; all patients received ongoing methylphenidate and parent training	12 weeks	Disruptive behavior as measured by NCBRF-D total score	No significant difference in NCBRF-D total score between groups.	4
Jahangard et al., 2017 [47]	Randomized, controlled, double-blinded	Children aged 7–10; ODD with comorbid ADHD; no history of intellectual disability (*n =* 84)	Adjunctive risperidone (0.5 mg/day) vs. placebo; all patients received ongoing methylphenidate (1 mg/kg/day)	8 weeks	ADHD and ODD symptoms, measured by CPRS-R subscale scores	Risperidone > placebo on CPRS-R scores for inattention, hyperactivity, and oppositional problems.	5
Juarez-Trevino et al., 2017 [48]	Randomized, controlled, double-blinded	Children and adolescents aged 6–16; CD with significant aggression; IQ > 70 (*n =* 24)	Risperidone (0.05 mg/kg/day) vs. clozapine (0.6 mg/kg/day)	16 weeks	Aggression as measured by MOAS total score	Risperidone = clozapine in terms of reduction in MOAS total score.	3
Masi et al., 2017 [49]	Naturalistic	Children and adolescents aged 6–16; ODD with comorbid ADHD; IQ > 70; drug-naïve (*n* = 40)	Risperidone (mean dose 1.5 mg/day) vs. methylphenidate (mean dose 20 mg/day); no concurrent treatment	6 months	ADHD and ODD symptoms, as measured by CBCL attention, rule-breaking, aggressive, ADHD, ODD and CD subscales	Risperidone = methylphenidate in terms of reduction in CBCL rule-breaking, aggressive, ODD and CD scores; methylphenidate > risperidone in CBCL attention and ADHD scores.	N/A
Blader et al., 2021 [50]	Randomized, controlled, double-blind	Children aged 6–12; DBD with comorbid ADHD and significant aggression; non-response to prior optimization of methylphenidate treatment (*n* = 45)	Risperidone (0.5–2.5 mg/day) vs. divalproex (375–1000 mg/day) vs. placebo; all patients received ongoing methylphenidate	8 weeks	Aggression, as measured by R-MOAS total score	Risperidone > divalproex and placebo in terms of reduction in R-MOAS total score	5

**Table 4 biomedicines-10-02818-t004:** Adverse drug reactions reported in recent clinical trials of antipsychotics for patients with disruptive behavior disorders.

Adverse Event	Frequency in Antipsychotic Group	Frequency in Control Group *
**Neurological**		
Dizziness	0–13%	22%
Extrapyramidal	0–8%	N/A
Headache	11–21%	44%
Insomnia	4–39%	67%
Nightmares	11%	22%
Sedation	6–17%	N/A
Somnolence	10–63%	11%
Syncope	8%	N/A
Tics	0–11%	33%
**Behavioural**		
Anxiety	6–33%	N/A
Apathy	22%	33%
Crying	39%	67%
Decreased speech	17%	22%
Depression	33%	56%
Euphoria	17%	22%
Increased speech	33%	78%
Irritability	58–61%	89%
Restlessness	56%	56%
**Digestive**		
Abdominal pain	6–13%	33%
Constipation	4–11%	0%
Decreased appetite	0–33%	89%
Dry mouth	6%	22%
Dyspepsia	11%	44%
Increased appetite	11–42%	22%
Nausea	13%	N/A
Sialorrhea	4%	N/A
**Respiratory**		
Cough	4%	N/A
Nasal congestion	4%	N/A
Rhinorrhea	6%	N/A
**Dermatological**		
Bruising	11%	22%
Rash	17%	11%
Genito-urinary		
Menstrual irregularity	6%	0%
Nocturnal enuresis	6–13%	11%
**Other/unspecified**		
Fatigue	11%	11%
Lack of energy	22%	44%

* Patients were receiving stimulant therapy (methylphenidate) with adjunctive placebo. N/A, data not available.

## Data Availability

Not applicable.

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
