# Peer review of "Antipsychotics in the Management of Disruptive Behavior Disorders in Children and Adolescents: An Update and Critical Review"

_biomedicines, 2022, doi:10.3390/biomedicines10112818_

Round 1

Reviewer 1 Report

Well-written review on the (off-label) use of neuroleptics in children and adolescents with DBD. This review describes in detail and accurately 6 recent studies on this topic. Conclusions and limitations are well presented and clinically very relevant.

Author Response

I thank the reviewer for their kind comments on my manuscript. I have made revisions and corrections in line with the suggestions of all reviewers and have corrected grammatical and spelling errors where these were present.

Reviewer 2 Report

This is a very interesting paper focused on the treatment of disruptive behavior disorders with antipsychotic medications. The paper is well-written and of interest for the journal.However, I propose several minor changes to be made before considering the paper for publication.

Abstract.

1-The introduction of the topic in the abstract section should be brief, and the main objectives should be described.

2- Are the authors conducting a non-systematic review? Please, provide search terms and some inclusion criteria. 

3-The study design of the six relevant trials should be described in detail. It is important. Are all the trials RCTs, open label, etc?

Introduction

1- I recommend a reference for the first sentence of the introduction section. Is there any other disorders to be potentially included in this group?

Material and methods.

1- The second section is mainly the results from systematic reviews...

I recommend to build a methods section to include the screening and selection processes. Databases that they used. Key-words, inclusion criteria etc. Although they conducted a non-systematic narrative review, the methods should be described.

2-How were the data extracted? If the authors conducted a qualitative synthesis, this should be also described.

Results.

1-The findings of the systematic reviews can be integrated as part of the discussion of the recent clinical trials. I consider that it has no sense to include systematic reviews in a separate section, and afterwards, the description of the clinical trials.

2-The main methods are described in section 3. It should be moved to a previous section. Subsections are needed.

3-Clinicaltrials. gov was searched to find addition studies. It should be integrated in the methods section and included in the flow diagram.

Author Response

I thank the reviewer for their thoughtful and in-depth critique of the original manuscript. I have made the following revisions in accordance with their suggestions:

The introduction of the topic in the abstract section should be brief, and the main objectives should be described.

Response: I agree with the reviewer. The abstract has been revised as follows:

"Disruptive behaviour disorders (DBDs) in childhood include conduct disorder (CD) and oppositional defiant disorder (ODD). Though psychological therapies are considered to be the first-line treatment for DBDs, many patients require adjunctive pharmacotherapy for the control of specific symptoms, such as aggression. Three prior systematic reviews have examined the evidence for the use of antipsychotics in DBDs and have concluded that their efficacy is marginal and limited by adverse effects. This paper has two objectives: (i) to summarize the findings of existing systematic reviews of antipsychotics for the management of DBDs in children and adolescents (2012-2017), and (ii) to provide an update to these reviews by examining recent clinical trials of antipsychotics in this population, published in the period 2-1-2017 to 2-10-2022."

 Are the authors conducting a non-systematic review? Please, provide search terms and some inclusion criteria.

Response: I agree with the reviewer. The revised version of the Abstract reads as follows: "The PubMed, Scopus and ScienceDirect databases were searched for relevant citations using the search terms “disruptive behaviour disorder”, “oppositional defiant disorder”, “conduct disorder” and their variants, along with “antipsychotic”, “atypical antipsychotic” and the generic names of all currently approved atypical antipsychotics. Trials were included only if they evaluated the efficacy of antipsychotics as monotherapy or adjunctive therapy for DBDs in children and adolescents."

The study design of the six relevant trials should be described in detail. It is important. Are all the trials RCTs, open label, etc?

Response: I apologize for this omission. This has been included in the revised Abstract as follows: "Six relevant trials were identified during this period, including five randomized controlled trials and one naturalistic open-label trial."

I recommend a reference for the first sentence of the introduction section. Is there any other disorders to be potentially included in this group?

Response: I thank the reviewer for this important suggestion. This has been corrected in the revised Introduction as follows:

"Disruptive behavior disorders of childhood and adolescence (DBDs) include oppositional defiant disorder (ODD) and conduct disorder (CD) [1-3]. Other conditions previously included in this group are attention-deficit/hyperactivity disorder (ADHD) and intermittent explosive disorder (IED). However, ADHD is now understood as a distinct disorder of neurodevelopmental origin [4], while IED, which has its onset in adolescence or adult life, is classified with the impulse control disorders [5, 6]. These latter conditions may be co-morbid with DBDs, but are considered to be distinct from them [7, 8]."

The second section is mainly the results from systematic reviews...

I recommend to build a methods section to include the screening and selection processes. Databases that they used. Key-words, inclusion criteria etc. Although they conducted a non-systematic narrative review, the methods should be described.

Response: I apologize for this error in the organization of the original manuscript. A new section covering all the review methods and the above points (Section 2) has been added, including a mention of databases, search terms, and inclusion criteria.

How were the data extracted? If the authors conducted a qualitative synthesis, this should be also described.

Response: I agree with the reviewer on the importance of these details. This has been mentioned in Section 2.4 of the revised manuscript as follows:

"2.4. Qualitative synthesis of included trials

For each trial, information was tabulated as follows: year of publication, study inclusion criteria including comorbid diagnoses permitted, sample size and age distribution, study design, study interventions (drug, dosage, and whether used as monotherapy or adjunctive therapy), and study results pertaining to efficacy and safety, including the frequency of reported adverse events.

For all randomized controlled trials, study quality was assessed using the Jadad scale [52], which is a well-established and valid method to assess the quality of reporting in such trials [53]. The overall results of these recent studies were then examined in the lights of the concerns identified in earlier systematic reviews."

The findings of the systematic reviews can be integrated as part of the discussion of the recent clinical trials. I consider that it has no sense to include systematic reviews in a separate section, and afterwards, the description of the clinical trials.

Response: I agree with the reviewer's suggestion regarding re-organization of the manuscript. The findings of the systematic reviews have now been included under "Results", just prior to the description of the trials, and have been placed after "Methods".

The main methods are described in section 3. It should be moved to a previous section. Subsections are needed.

Response: I agree with the reviewer. The Methods have been shifted to a separate section (Section 2) prior to discussion of the results. This section has been divided into four subsections as follows:

2.1 Identification and summarization of existing systematic reviews

2.2 Identification of recent clinical trials

2.3 Identification of unpublished trial data

2.4 Qualitative synthesis of included trials

All methodological details have been moved from "Results" to this section.

Clinicaltrials. gov was searched to find addition studies. It should be integrated in the methods section and included in the flow diagram.

Response: I agree with the reviewer. This has been added to the Methods section as follows:

"2.3. Identification of unpublished trial data

As earlier systematic reviews had identified relevant results from an unpublished trial, the ClinicalTrials.gov database was searched to examine if any unpublished trials of relevance could be identified for the period 2017-22, using the diagnostic categories “Disruptive Behavior Disorder” and “Childhood Disruptive Behavior Disorder” (provided by the database’s classificatory system) along with “antipsychotic”, “antipsychotics”, both alone and in combination with “typical” or “atypical” as well as the list of antipsychotics mentioned above. Of 27 records retrieved through this search, only one study of potential relevance was identified in the time period 2017-22. This trial (identifier: NCT02063945) was a head-to-head, open-label comparison of risperidone and methylphenidate in children and adolescents with DBDs and comorbid ADHD. Only five subjects were recruited for the study, and it was prematurely terminated as of 2020 due to difficulty in recruiting participants; no results have been posted to date. Therefore, it was not possible to include the detailed results of this abandoned trial in the current review [51]."

The flow diagram (Figure 1) has also been edited to include this detail.

Reviewer 3 Report

The introduction will be benefit from discussion and referencing the work of Professor Saumitra Deb et al who has done extensive work in this area, there is also work by Tyrer et al from the University of Oxford, and it will be useful to reference the UK NICE Guidelines on the prescribing of psychotropic medicines in people with learning disability. These guidelines apply across settings.

Author Response

I thank the reviewer for identifying certain important lacunae in the literature review of the original manuscript, and have made corrections as follows:

The introduction will be benefit from discussion and referencing the work of Professor Saumitra Deb et al who has done extensive work in this area

Response: I apologize for omitting references to the work of Prof. Deb, who has carried out important research in this field. New references [36-39] to this work have been added and discussed in the Introduction:

"Subsequent trials suggested that these drugs were not superior to placebo in the management of DBD in patients with comorbid ID [36-38], and experts in the fields highlighted the absence of supporting evidence for the use of these drugs in DBDs, as well as the ethical issues associated with their widespread prescription [39, 40]."

There is also work by Tyrer et al from the University of Oxford.

Response: I agree with the reviewer. References to the work of Prof. Tyrer have been added [36, 39, 40] and discussed in the above paragraph. They have also been cited in the Discussion section where relevant.

It will be useful to reference the UK NICE Guidelines on the prescribing of psychotropic medicines in people with learning disability. These guidelines apply across settings.

Response: I agree with this suggestion by the reviewer. This important guideline, which addresses many of the same concerns as the current paper, has been cited [55] and discussed in the Discussion as follows:

"Standard practice guidelines, such as those published by the United Kingdom’s National Institute for Health and Care Excellence, take a cautious approach to the prescription of antipsychotics for children and adolescents with DBDs, with their use confined to short-term treatment of severe aggression resistant to psychosocial interventions [55]. Despite such recommendations, the use of atypical antipsychotics in children and adolescents with DBDs has increased markedly in the past two decades [32-34, 44, 56]."